# The Brinell Method for Determining Hardness of Wood Flooring Materials

**Maciej Sydor \*** , **Grzegorz Pinkowski** and **Anna Jasińska**

Faculty of Wood Technology, Poznań University of Life Sciences, 60-637 Poznań, Poland; grzegorz.pinkowski@up.poznan.pl (G.P.); anna.jasinska@up.poznan.pl (A.J.)

\* Correspondence: maciej.sydor@up.poznan.pl

**Abstract:** We hypothesize that the ability to recovery the depth of the indentation increases with increasing the hardness of the flooring material. The research was carried out for ten lignocellulosic flooring materials: merbau, oak, maple, red oak, laminated HDF (high-density fiberboard), innovative plywood, beech, pine, peasantry, iroko. The hardness was examined using the Brinell method, and additionally, the elastic indentation of the indenter was measured during the hardness test. On this basis, the permanent (plastic) and temporary (elastic) component of total deformation was determined. Different ability to recovery was found. The harder materials were the higher percentage of elastic indentation in total indentation depth. Moreover, it was found that the measurement of the indentation diameter in wood materials is characterized by high uncertainty and measurements based on the depth of the indentation are more unambiguous and of greater practical importance, especially when testing hard lignocellulosic flooring materials.

**Keywords:** Brinell hardness; flooring materials; indentation depth; plastic deformation; elastic deformation; shape recovery

---

## 1. Introduction

The hardness is a crucial wood mechanical propriety, mainly because it positively and negatively correlates with density and moisture content [1]. It also depends on the anatomical direction and can vary by up to 50% within the same species [2]. Janka and Brinell are the two most popular methods for determining hardness of wood materials [3,4]. Schwab [5] compared different hardness methods for 16 wood species and concluded that the Brinell hardness test of flooring materials obtained the most reliable results.

For Brinell measurements, a steel or carbide ball is pressed into the sample, the ball is removed, and the diameter of the resulting permanent indentation is measured using a microscope. The largest indentation of the ball occurs during the measurement (under the measuring force), while when the force is removed, this indentation automatically decreases (elastic partial recovery). Since the different types of wood vary considerably in their strength properties (and especially in their stiffness), different percentages of permanent plastic deformation in the total deformation caused by the measuring ball should be expected. This has been noticed by many researchers proposing to modify the Brinell method to measure the depth of the indentation under load during the test and to calculate the hardness on this basis [6–12]. Measuring the depth of the indentation under load, instead of measuring the diameter of the plastic indentation visually, provides Brinell hardness values better correlated with the density of measured wooden samples [12]. This approach is very well suited for measuring resilient wooden samples [8,10,12], and especially for measuring the performance of flooring materials [6], because the measurement of a total indentation of a measuring ball, taking into account both plastic and elastic deformation, better describes the performance of the wood, compared to measuring only

the plastic indentation as in the classic Brinell method. High hardness is important in flooring materials and that no permanent deformation occurs under the influence of loads concentrated on a small area.

Research hypothesis: The ability to recover the depth of the indentation after removing the load increases with increasing the hardness of the flooring material. In order to verify the hypothesis, we compared the ability to self-disappearance of the indentation for different wood materials.

## 2. Materials and Methods

### 2.1. Materials

The research described in this article concerned six different flooring materials and four types of wood. The experiments were carried out on samples of different structures: two-layer typical floor panels made of different wood species (samples from A to D), single-layer typical high-density fiberboard (HDF) floor panels (sample E), double-layer floor panels made of vertical birch plywood with a rolled-up structured surface of top layer (sample F), and single-layer samples made of selected wood species (samples G to J). All two-layered samples had the lower layer made of spruce wood. The determinations, material names of layer thickness, and density of test samples are presented in Table 1.

**Table 1.** Characteristics of test samples.

| Name | Latin Name | Specimen Designation | Thickness (mm) | Density (g/cm$^3$) | Average Density (g/cm$^3$) | Decription |
|---|---|---|---|---|---|---|
| Merbau | *Intsia bijuga* (Colebr.) Kuntze | A | 3.33 | 0.69 | 0.56 | Top layer |
| Common spruce | *Picea abies* L. | | 7.77 | 0.50 | | Bottom layer |
| Pedunculate oak | *Quercus robur* L. | B | 3.34 | 0.64 | 0.54 | Top layer |
| Common spruce | *Picea abies* L. | | 7.66 | 0.50 | | Bottom layer |
| Maple | *Acer saccharum* Marsh. | C | 3.33 | 0.68 | 0.55 | Top layer |
| Common spruce | *Picea abies* L. | | 7.74 | 0.50 | | Bottom layer |
| Red oak | *Quercus rubra* L. | D | 3.19 | 0.77 | 0.58 | Top layer |
| Common spruce | *Picea abies* L. | | 8.06 | 0.50 | | Bottom layer |
| HDF | - | E | 6.73 | - | 0.94 | One layer |
| Vertical birch plywood | - | F | 3.60 | 0.69 | 0.56 | Top layer |
| Common spruce | *Picea abies* L. | | 7.50 | 0.50 | | Bottom layer |
| Common beech | *Fagus sylvatica* L. | G | - | - | 0.68 | One layer |
| Pine | *Pinus sylvestris* L. | H | - | - | 0.56 | One layer |
| Black locust | *Robinia pseudoacacia* L. | I | - | - | 0.69 | One layer |
| Iroko | *Milicia excelsa* (Welw.) CC Berg | J | - | - | 0.50 | One layer |

All samples were stored in the laboratory under the same temperature (av. 20 °C) and humidity conditions and were air-conditioned to the equilibrium moisture content (humidity in the range of 5.0–6.5% was determined by oven dry test).

Samples A, B, C, D, and F are two-layer. Samples E, G, H, I, J are single-layer (as described in Table 1). Sample H (HDF) was made in a conventional way for this type of material. Additionally, it was covered with a layer consisting of decorative paper impregnated with melamine.

Innovative material F is a kind of birch plywood with vertical sheets. Its anatomical grain directions of fibers in neighboring veneers form a 90-degree angle (typical for plywood), but its veneers are arranged vertically (differently from typical plywood). As a result of such a structure, the top floor layer consists of alternately with radial and tangential sections. This new material has the trade name Studio Loft$^{TM}$. The pressing pressure of the vertical plywood block is about 1.5 MPa and there is no heating plate used here. Gluing takes place at the temperature prevailing in the production hall (approx. 20 °C), pressing time is about 20 min, and the glue used is a standard PVAc glue (type C3). After pressing, the blocks of plywood are cut into layers with a thickness of 4.2 mm. The top side of

layers are structured (embossed with a hot roller). The spruce supporting material (bottom panel layer) is glued in the same way as "vertical plywood blocks" (cold with PVAc glue, pressing time 20 min).

Figure 1 shows views of samples made of double-layered materials (samples from A to D, and sample of innovative flooring material F).

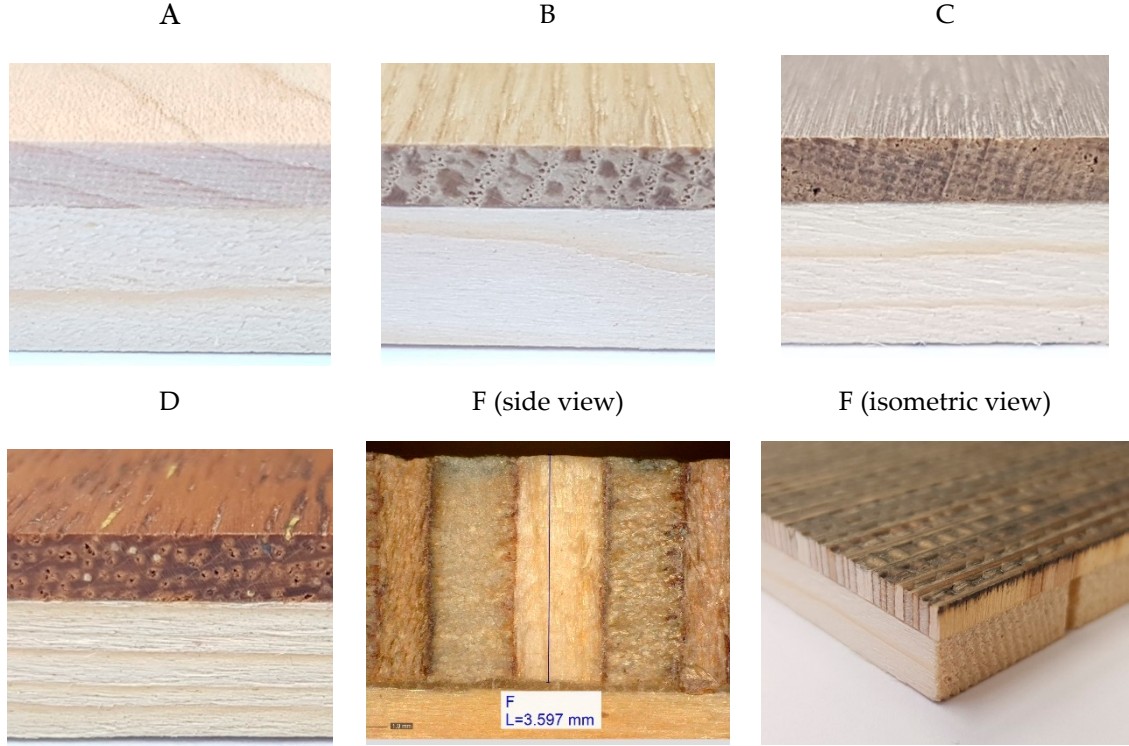

**Figure 1.** Two-layered test materials (markings according to Table 1, photo GP).

## 2.2. Methods

Hardness was measured by the Brinell method using HBRV-187.5E hardness tester (HUATEC Group, Beijing, China) (Figure 2). The following parameters were used during the tests:

| | |
|---|---|
| Ball diameter | $D = 10.0$ mm |
| Total load | $P = 30.0$ kG ($F = 294.2$ N, ($F/D^2 = 2.9$) |
| Partial load | $P_1 = 10.0$ kG (98.07 N) |
| Total load time | $t = 60$ s |
| Number of measurements for each material | $n = 12$ |
| Symbolic specification of test conditions | $HB_D \rightarrow$ HB 10/294.2/60. |

The application of load used was different from that required by EN 1534 (1 kN reached in $15 \pm 3$ s, maintained for $25 \pm 5$ s and entirely released [13]). In the Brinell method, the diameter of the indentation should be in the $0.25D \leq d \leq 0.6D$ range (where $D$ is the indentation diameter). Therefore, a lower force than that recommended in the standard EN 1354 was used. As a result, the diameter of the indentation in the tests ranged from 2.29 mm for sample F (innovative plywood) to 4.97 mm for sample H (pine). In addition, the time of force exertion was lengthened to reduce the elastic component of deformation.

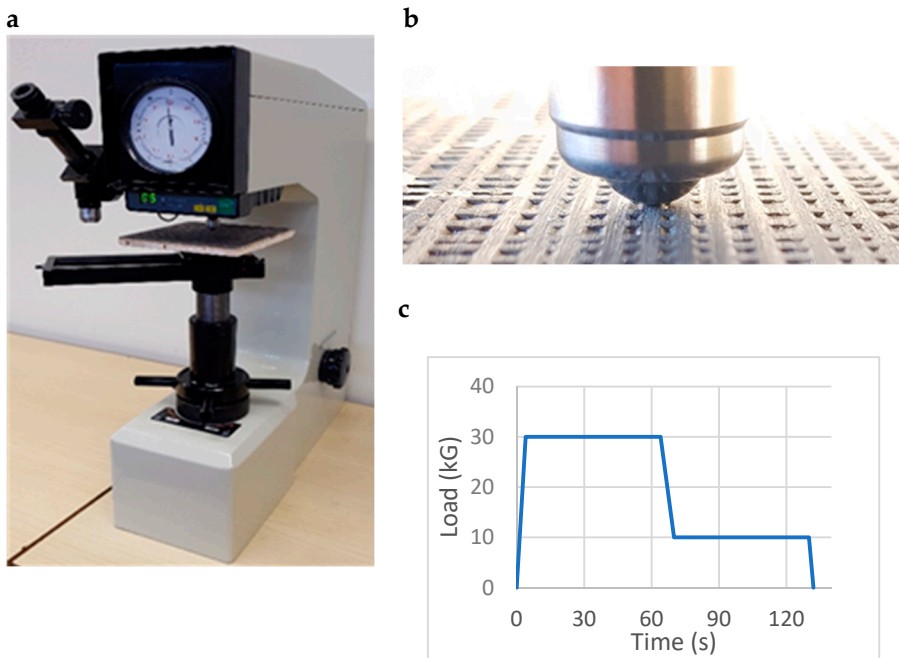

**Figure 2.** Test stand (photo GP): (**a**) hardness tester HBRV-187.5E, (**b**) indenter, steel ball with a diameter of 10 mm, (**c**) load exertion mode.

The force on the indenter was exerted as follows: for 4 s, the force was increased from 0 to P, for another 60 s, the force P was maintained, for another 6 s, the force was reduced to P1 for another 60 s the load P1 was maintained, and then the sample was completely relieved. The nature of load changes is shown in Figure 2c.

After the sample was unloaded, 3 min was waited before measuring the indentation diameter. For wood materials, there are difficulties in clearly defining the border of the indentation. Measuring the diameter of an indentation with a microscope is subjective, because the border of an indentation in wood materials is not clear, and it is up to the researcher to determine the actual border of an indentation correctly [11,14]. An additional complication is the "sinking-in effect" [15], occurring especially when force is applied in radial or in tangential direction [12]. Therefore, the Dino-Lite AM4815ZT EDGE digital microscope (manufactured by IDCP B.V., Almere, Netherlands) with additional digital functions such as EDR (extended dynamic range) and EDOF (extended depth of field) and the possibility of measuring under polarized light (which made it even easier to find the limits of the indentation). The enlargement of the image was matched to the diameter of the indentation received. Example images of indentations created during hardness measurement are shown in Figure 3 (the photographs were taken at 37× magnification).

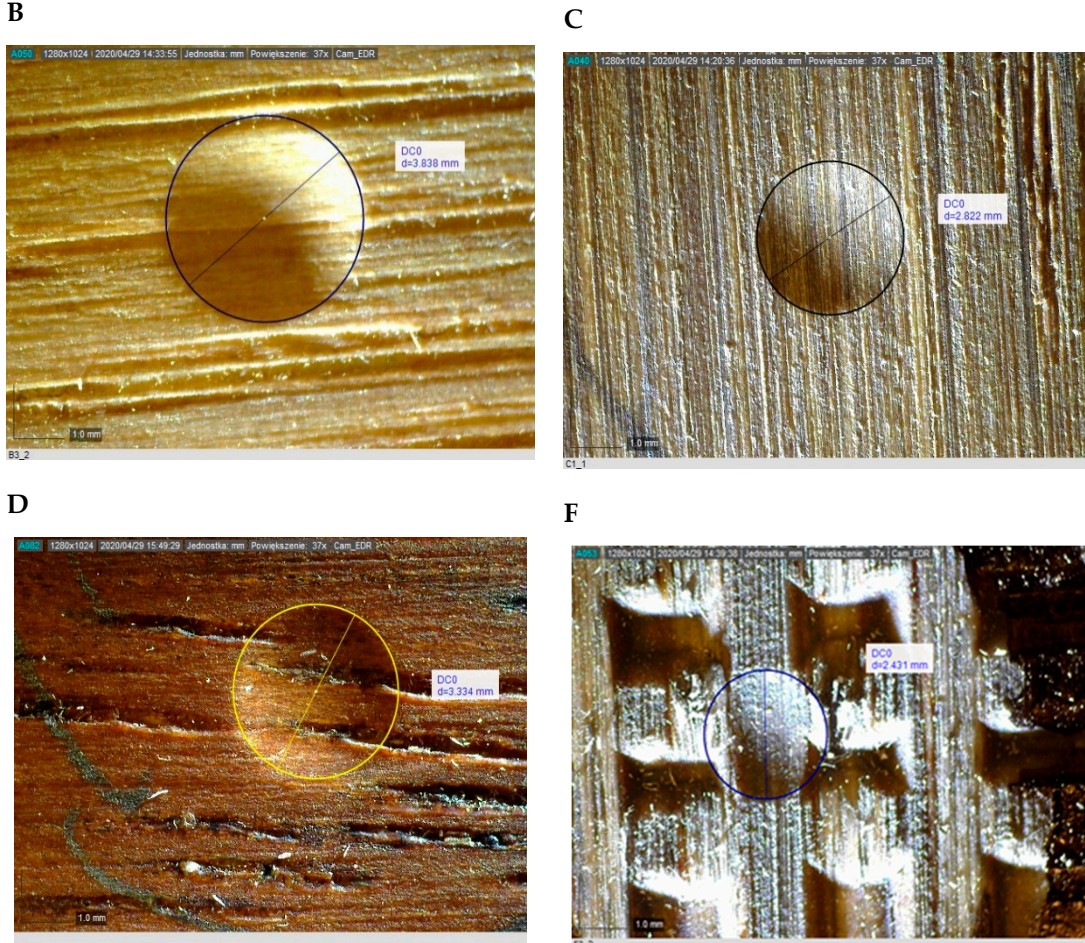

**Figure 3.** The indentations in the selected samples (markings according to Table 1): B (pedunculate oak), C (maple), D (red oak) and F (innovative plywood) (photo GP).

The hardness of the tested samples was calculated in two ways, based on the diameter of the plastic imprint of the ball in the sample (measurement with a microscope, after the load of the sample, which is typical for the Brinell method) and on the total depth of the indentation (measurement with a sensor in the hardness meter, indentation under the load of the measuring force). The Brinell hardness, calculated from the diameter of the plastic indentation, was marked as $HB_d$ and calculated according to a Formula (1):

$$HB_d = \frac{2 \cdot P}{\pi \cdot D \cdot \left(D - \sqrt{D^2 - d^2}\right)},$$ (1)

where:

$P$ = applied load in kilogram-force (kGf);
$D$ = diameter of indenter (mm);
$d$ = diameter of indentation (mm).

The total hardness, calculated from the total depth of the ball indentation loaded with the measuring force, was marked as $HB_H$ and calculated according to the Formula (2):

$$HB_H = \frac{P}{\pi \cdot D \cdot H},$$ (2)

where:

$H$ = depth of imprint under load (mm).

HB$_\text{d}$ values include only plastic deformation (Figure 4b), while HB$_\text{H}$ values include both elastic and plastic deformation (Figure 4a). This is due to the natural elasticity of wood plastics, as the ball's compression depth H of the loaded ball P is greater than the ball's compression depth H when the load is removed. The hardness tester enables this difference to be measured, it is $x = H - h$ and is a measure of the flooring material's ability to automatically disappear the indentation crushed by the ball when the load is removed (recovery).

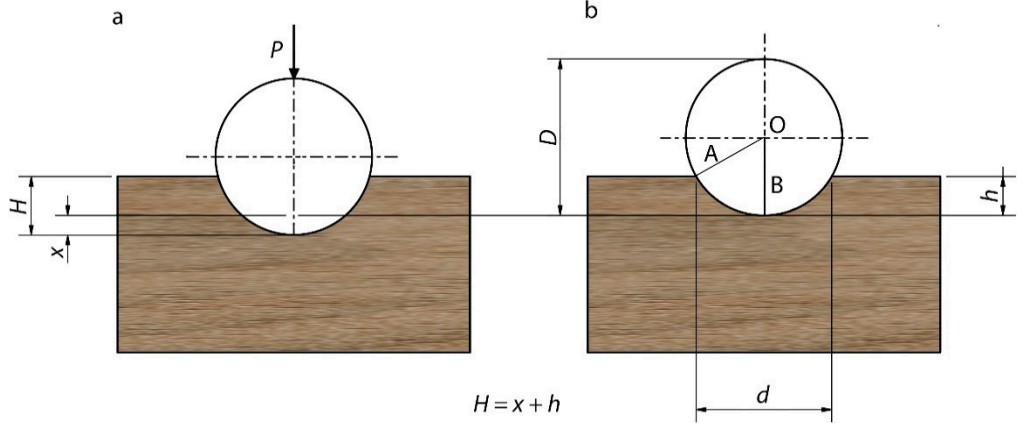

**Figure 4.** Geometrical relationships when measuring hardness using the Brinell method: (**a**) ball under load, (**b**) ball after load is removed (*P*—measuring load, *D*—diameter of the ball, *d*—diameter of the indentation, *H*—total indentation, *h*—plastic indentation (permanent), *x*—elastic indentation (temporary)).

In order to find the recoverability, a relationship was derived to calculate the value of *h*. This was done based on ball diameter *D* and measured values of *d* and *x*. For the rectangular triangle, AOB (Figure 4) can be recorded:

$$x = \sqrt{\left(\frac{D}{2}\right)^2 - \left(\frac{d}{2}\right)^2}. \tag{3}$$

Since $h = H - x$ (Figure 4), then *h* can be calculated using a relationship:

$$h = \frac{D}{2} - \sqrt{\left(\frac{D}{2}\right)^2 - \left(\frac{d}{2}\right)^2}. \tag{4}$$

The diameter of the indentation *d* (with a microscope) and the elastic component of the indentation *x* (with a hardness meter) were measured. The following were calculated: *h* (based on *d* and *D*), *H* (based on *h* and *x*), Brinell hardness HB$_\text{d}$ (based on *P*, *D* and *d*), and total hardness HB$_\text{H}$ (based on *P*, *D*, and *H*).

## 3. Results

Figure 5 shows the calculated hardness of HB$_\text{d}$ and HB$_\text{H}$ of the tested samples together with error bars (estimated for *n* = 12, Student distribution, confidence interval 90%).

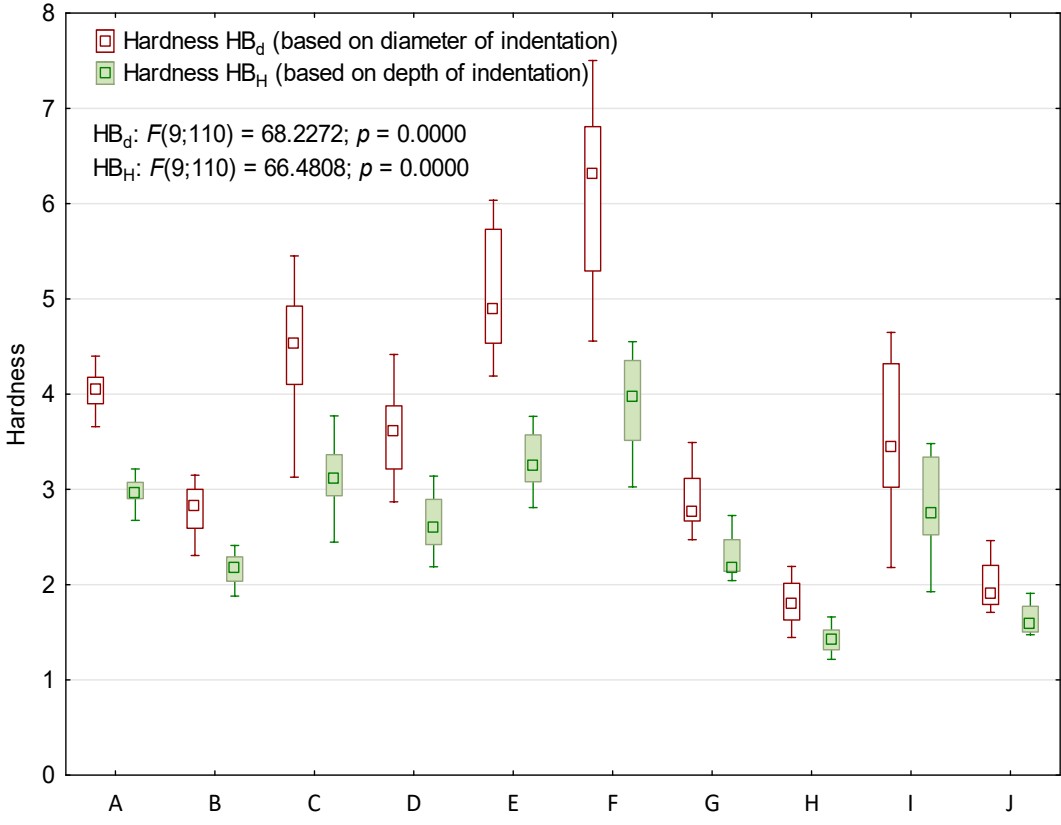

**Figure 5.** Average hardness values for the tested materials.

The hardness values shown in Figure 5 are listed in Table 2, additionally giving the percentage difference between the two hardness values.

**Table 2.** Results of hardness measurements based on indentation diameter ($HB_d$) and indentation depth ($HB_H$) and homogeneous groups.

| Genus | Specimen Designation | $Hb_d$ | HG | $HB_H$ | HG | The Difference Between the Determined Hardness Values ($1-HB_H/Hb_d$) |
|---|---|---|---|---|---|---|
| Merbau | A | 4.04 ± 0.11 | d | 2.98 ± 0.08 | de | 26% |
| Oak 1 | B | 2.84 ± 0.20 | b | 2.18 ± 0.12 | b | 23% |
| Maple | C | 4.44 ± 0.37 | d | 3.12 ± 0.21 | ef | 30% |
| Oak 2 | D | 3.56 ± 0.25 | c | 2.64 ± 0.15 | c | 26% |
| HDF | E | 5.06 ± 0.35 | e | 3.30 ± 0.16 | f | 35% |
| Plywood+ | F | 6.06 ± 0.47 | f | 3.87 ± 0.27 | g | 36% |
| Beech | G | 2.88 ± 0.17 | b | 2.29 ± 0.12 | b | 20% |
| Pine | H | 1.81 ± 0.12 | a | 1.43 ± 0.07 | a | 21% |
| Robinia | I | 3.53 ± 0.44 | c | 2.80 ± 0.28 | cd | 21% |
| Iroko | J | 1.99 ± 0.14 | a | 1.64 ± 0.09 | a | 18% |

The results given in Table 2 show that the $HB_H$ hardness of the materials tested is always lower than the $HB_d$ hardness. From the point of view of the differences found between $HB_d$ and $HB_H$, the materials tested can be divided into three groups. The difference ranges from about 18–21% for wood samples G, H, I, J, through about 23–30% for parquet wood (A, B, C, D), up to 35–36% for pressed wood-based materials (E, F).

The results of the ANOVA analysis of variance, with a significance level of 0.05, show statistically significant differences between the mean values of hardness, depending on the type of sample. Table 2

shows the division of average hardness measurements into homogeneous groups (HG). In the case of $HB_d$ hardness, no differences between the average values occurred for four pairs of samples, i.e., (H, J), (B, G), (D, I), and (A, C). Samples E and F differed from all others. In the case of $HB_H$ hardness, the division was analogous to that for $HB_d$, with the difference that for samples A, C, D, I, double homogeneous groups were determined.

Statistical inference was performed with the use of Statistica 13 software [16]. The analysis of variance (ANOVA) and regression analysis were performed for $HB_d$ and $HB_H$ hardness. The significance level of $p = 0.05$ was adopted for both analyses (Table 3). Duncan's test was used to find statistically significant differences between the mean values in the analysis of variance. The $H_0$ hypothesis was tested, assuming that the model is not statistically significant, assuming the level of $p = 0.05$. The significance level for Fisher's test was determined and the obtained $p < 0.05$. The obtained $p$ values and the results of the F test are given in Table 3.

**Table 3.** The analysis of variance (ANOVA) results.

| $HB_d$ | | | | |
|---|---|---|---|---|
| | **Sum of Squares SS** | **Degrees of Freedom DOF** | **Mean Squares MS** | **Fisher's F-Test** | **Sig. Level $p$** |
| Intercept | 1576.152 | 1 | 1576.152 | 4986.932 | 0.00 |
| Type of sample | 194.073 | 9 | 21.564 | 68.227 | 0.00 |
| Error | 34.766 | 110 | 0.316 | | |
| Total | 228.839 | | | | |

| $HB_H$ | | | | |
|---|---|---|---|---|
| **Effect** | **Sum of Squares SS** | **Degrees of Freedom DOF** | **Mean Squares MS** | **Fisher's F-Test** | **Sig. Level $p$** |
| Intercept | 830.8346 | 1 | 830.8346 | 7859.184 | 0.00 |
| Type of sample | 63.2522 | 9 | 7.0280 | 66.481 | 0.00 |
| Error | 11.6287 | 110 | 0.1057 | | |
| Total | 74.8809 | | | | |

The values of total deformations under measurement load are presented in Figure 6 as a cumulative graph. This graph shows the total deformation divided into plastic and elastic components (the graph also has error bars: $n = 12$, Student distribution, confidence interval 90%).

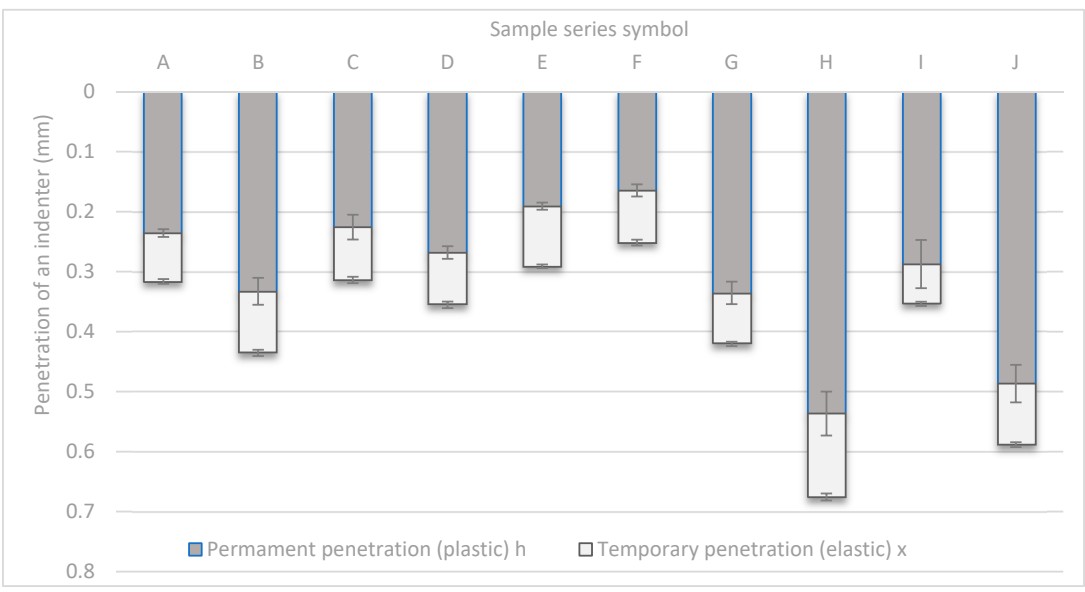

**Figure 6.** Total deformation of samples under load.

Figure 6 shows that for the used load conditions, the smallest values of total indentation were measured for samples F (pressed plywood) and the largest for samples H (pine). The measured and calculated deformation values are presented in Table 4, which also includes the calculated share of permanent (plastic) indentation in the total indentation depth.

**Table 4.** Indentations: total, spring, plastic, and share of spring indentation in total indentation.

| Genus | Specimen Designation | Total Indentation $H$ (mm) | Spring Indentation $X$ (mm) | Plastic Indentation $H$ (mm) | The Share of A Plastic Indentation ($H/H$) |
|---|---|---|---|---|---|
| Merbau | A | 0.32 | 0.08 | 0.24 | 75% |
| Common oak | B | 0.44 | 0.10 | 0.33 | 76% |
| Maple | C | 0.31 | 0.09 | 0.23 | 72% |
| Red oak | D | 0.36 | 0.09 | 0.27 | 75% |
| HDF | E | 0.29 | 0.10 | 0.19 | 65% |
| Plywood+ | F | 0.25 | 0.09 | 0.16 | 65% |
| Beech | G | 0.42 | 0.08 | 0.34 | 80% |
| Pine | H | 0.68 | 0.14 | 0.54 | 79% |
| Robinia | I | 0.35 | 0.07 | 0.29 | 81% |
| Iroko | J | 0.59 | 0.10 | 0.49 | 83% |

From Table 3, it is possible to read that the smallest value of the elastic component and the smallest value of the permanent component in the total indentation have the samples F (pressed plywood), whereas the highest values–samples H. The share of plastic indentation in the total indentation ($h/H$) is the smallest for the samples F and is 65%, and the largest for the samples J and is 83%.

Variability of the indentation depth over time during the test is shown in Figure 7. This variability corresponds to the course of the load measurement force shown in Figure 2c. The force loading the ball increased from 0 to 30 kG in the first 4 s, then it was 30 kG for 60 s, to decrease to 10 kG in the next 6 s and to zero in the next 60 s.

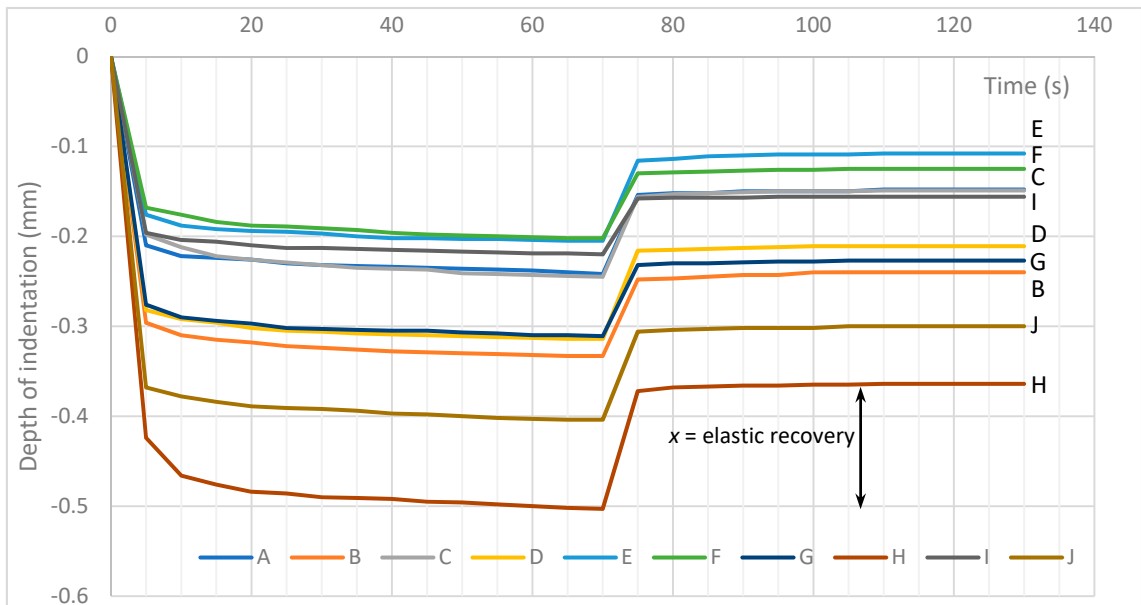

**Figure 7.** Variability of the indentation depth during the test.

Figure 8 shows the relationship between hardness and density: $HB_d$ (calculated from the indentation diameter) and $HB_H$ (calculated from the indentation depth). For this relationship, a regression analysis was performed, for which trend lines, straight line equations, correlation and determination coefficients were determined. Based on the regression analysis, it can be concluded that the regression model and correlation coefficients are statistically significant. The analysis did not include data on the F.

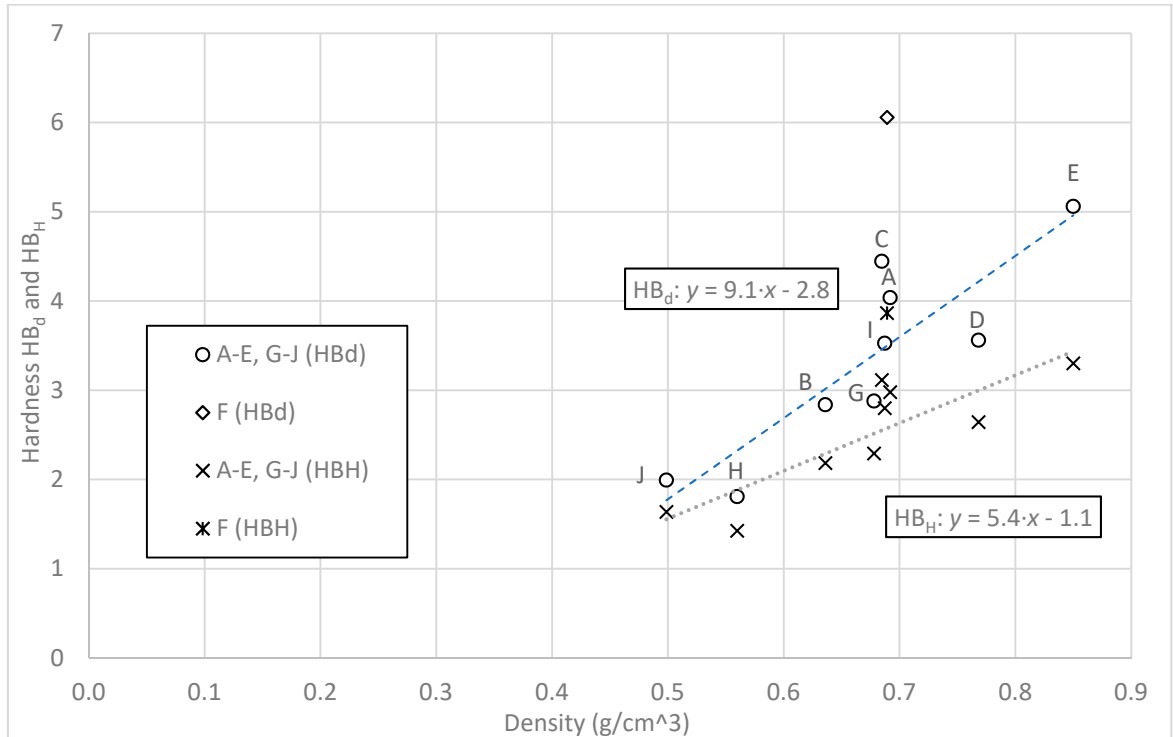

**Figure 8.** Hardness of $HB_d$ and $HB_H$ depending on sample density.

Table 5 summarizes the results of the regression analysis. The correlation and determination coefficients show that the obtained data fit the regression model adopted. Their numerical values are relatively low, which results from the large dispersion of the measured hardness values and probably results from the wide variety of properties of the samples.

**Table 5.** Results of the regression analysis for hardness versus density.

| Hardness | $HB_d$ | $HB_H$ |
|---|---|---|
| Regression equation | $y = -27{,}646 + 90{,}888 \cdot x$ | $y = -11{,}155 + 53{,}558 \cdot x$ |
| Correlation coefficient | 0.7846 | 0.7698 |
| The coefficient of determination $R^2$ | 0.6156 | 0.5926 |
| Significance level $p$ | 0.0000 | 0.0000 |

For samples with low density, small values of both hardness ($HB_d$ and $HB_H$) can be observed, while as the density of the samples increases, the measured hardness increase. These correlations are roughly straightforward, as shown by the two trend lines. Due to the clear outlying value of the hardness of sample F, it was decided to exclude it from both trend lines. The determination coefficients of both trend lines show a relationship between the parameters analyzed. A comparison of the directional coefficients of the trend lines indicates that $HB_d$ hardness increases faster than $HB_H$ hardness.

## 4. Discussion

The comparison of hardness measurement results with those available in the literature is shown in Table 6. The cited authors used different values of the measuring force (120, 500, and 1000 N) and one measuring ball diameter (10 mm). The results of our research and the results from the literature have been converted into kG into square millimeters.

**Table 6.** Comparison of the tests results with literature.

| Genus | Specimen Designation | $HB_d$ (kG/mm$^2$) | $HB_H$ (kG/mm$^2$) | Literature |
|---|---|---|---|---|
| Merbau | A | 4.04 | 2.98 | $HB_d$ (10/500/30) = 33.2 MPa ≈ 3.42 kG/mm$^2$ [5] |
| Common oak | B | 2.84 | 2.18 | $HB_d$ (10/500/30) = 26.2 MPa ≈ 2.67 kG/mm$^2$ [5] |
| Maple | C | 4.44 | 3.12 | $HB_d$ (10/1000/nd) = 31–42 MPa ≈ 3.16–4.28 kG/mm$^2$ [17] |
| Red oak | D | 3.56 | 2.64 | $HB_d$ (5/120/nd) = 38 MPa ≈ 3.87 kG/mm$^2$ [18] |
| HDF | E | 5.06 | 3.30 | $HB_d$ (10/500/30) = 48.9 MPa ≈ 4.98 kG/mm$^2$ [19] |
| Plywood+ | F | 6.06 | 3.87 | - |
| Beech | G | 2.88 | 2.29 | $HB_d$ (10/500/30) = 26.7 MPa ≈ 2.72 kG/mm$^2$ [5] |
| Pine | H | 1.81 | 1.43 | $HB_H$ (10/1000/25) = 13 MPa ≈ 1.33 kG/mm$^2$ [11] |
| Robinia | I | 3.53 | 2.80 | $HB_d$ (nd.) = 37 MPa ≈ 3.77 kG/mm$^2$ [20] after [21] |
| Iroko | J | 1.99 | 1.64 | - |

As can be seen in Table 6, our hardness results are similar to the results reported in the literature cited.

As indicated in the introduction, the phenomenon of elasticity of wood materials during Brinell hardness testing was analyzed in the scientific literature. Kontinen and Nyman [22] compared two different ways of determining Brinell hardness for flooring materials. First, they calculated the hardness using the depth of the indentation in the sample. Then, the calculations were repeated using the diameter of the indentation. The hardness values obtained by measuring the diameter were about 60–160% higher than those obtained by measuring the indentation depth (a force of 1 kN and a 10 mm ball were used). The results described in this article also showed differences in the hardness calculated using two methods, but it was in the range of 18–36%. This difference can be explained by the use of a lower force in our hardness tests (0.29 kN vs. 1 kN). In our studies, elastic recovery for pine was 21% using a force of 292 N (Table 2). Similar results were obtained by Laine, Rautkari, and Hughes [11] (26%, using 1000 N).

Due the differences in hardness, measured by the diameter of the indentation and by the depth of the indentation, Kontinen and Nyman [22] concluded that measuring the depth of the indentation is a better and more precise method than measuring the diameter of the indentation. Similar conclusions were published by [6,12,23]. The method for calculating the Brinell wood hardness based on the depth of the indentation is also used in the Japanese standard JIS Z 2101:1994 [7].

The results show that hardness is correlated with the density of lignocellulose materials. This is consistent with the results reported by other researchers [24]. However, the coefficient $R^2$ for the $HB_d$ trend line takes higher values than the analogous coefficient for $HB_H$, which indicates a better correlation of $HB_d$ hardness with wood density than the correlation of $HB_H$ hardness (Figure 8). This is the opposite result to earlier experiments results [12]. Another new observation is that the "plastic" hardness determined from the indentation diameter ($HB_d$) increases faster, with increasing density, than the "total" hardness calculated from the indentation depth ($HB_H$).

## 5. Conclusions

Based on the research results, it can be concluded that the research hypothesis was confirmed (the ability to automatically reducing the depth of the indentation after removing the load in the Brinell method increases with the increasing hardness of the floor material). On the basis of the analysis of the obtained results, two main conclusions can be drawn.

- The materials tested in terms of hardness can be divided into three groups: soft materials (G, H, I, J—beech, pine, peasantry, iroko), intermediate materials (A, B, C, D—merbau, common oak, maple, red oak), and very hard materials (E, F—HDF, plywood+). In soft materials, the highest percentage of plastic indentation in total deformation was observed (79–83%), in intermediate materials this percentage is 72–76%, and in hard materials 65%. Thus, hard materials show the highest ability, among the materials tested, to reduction of the depth of deformation automatically after load removal.

- The hardness measured from the indentation diameter ($HB_d$) has, in each test case, a higher value than the hardness determined from the indentation depth ($HB_H$). This is intuitive. However, this difference is not constant and ranges from about 18–21% for soft materials (samples G, H, I, J), through about 23–30% for medium-hard materials (samples A, B, C, D), up to 35–36% for hard pressed wood-based materials (samples E, F). Different plastic and elastic component shares in total deformation are the result of different density of the tested lignocellulosic materials.

It can be also stated:

- The measurement of the depth of the indentation is much faster to make and, above all, more unambiguous than the measurement of the indentation diameter. In addition, the "visual" measurement of the diameter of a ball's indentation is subjective, as there is no clear border of the indentation in the case of wood materials. The depth of the indentation is determined unequivocally and with great accuracy, as it is based on the indications of the length sensor. It can be concluded that measuring the depth of the indentation gives more reliable hardness values.

- As the density of wood materials increases, their HB hardness also increases. This increase is linear in nature. The large scattering of hardness results may be since measurements were made on different (non-oriented) surfaces of both radial, tangential, and intermediate cross-sections. The significantly higher hardness values of sample F (plywood+) are due to the specific orientation of the wood fibers in this sample, causing hardness measurements to be carried out on the cross-sectional area of the veneer layers. This anatomical orientation of the veneers of the flooring material translates favorably into higher hardness of the floor made of it.

- Statistical analyzes revealed differences in hardness between the samples, but not in all cases, and the lack of differences between the samples is not necessarily related to the density of the sample.

**Author Contributions:** Conceptualization, M.S.; Methodology, G.P. and M.S.; Software, G.P.; Validation, M.S.; Formal Analysis, M.S.; Investigation, G.P.; Resources, M.S.; Data Curation, A.J. and G.P.; Writing—Original Draft Preparation, M.S.; Writing—Review & Editing, M.S.; Visualization, A.J. and G.P.; Supervision, M.S.; Project Administration, M.S.; Funding Acquisition, M.S.; Approximate percentage share in the article: M.S. 60%, G.P. 30%, A.J. 10%. All authors have read and agreed to the published version of the manuscript.

**Funding:** Financed within the framework of Ministry of Science and Higher Education program 'Regional Initiative of Excellence' in years 2019–2022, Project No. 005/RID/2018/19.

**Acknowledgments:** The authors thank Marcin Kowalski (Holzexport sp. o.o., Poland) for the free delivery of most of the test samples (samples A–F).

**Conflicts of Interest:** The authors declare no conflict of interest.

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
