# Peer review of "The Brinell Method for Determining Hardness of Wood Flooring Materials"

_forests, doi:10.3390/f11080878_

Round 1

Reviewer 1 Report

  • line 27: It is such an unusual formulation, instead of "[5] He" it should be put there that Schwab [5] compared different methods ...
  • lines 64-65: the second parenthesis is missing ")"
  • line 67: There is no completed sentence - and ... what?
  • lines 53-55: please describe in more details the construction of both composites: 1. about "single-layer HDF floor panels" we know its thickness and density, but we do not know the type of HDF surface treatment, the predominant type of wood (woods) in its structure and the method of its pressing due to the possible higher density in the surface layer (density profile); 2. about "double-layer floor panels made of innovative laminated wood materials with a rolled-up structured surface layer under the trade name Studio loftTM" we suspect that birch veneer was pressed with spruce veneer, but what is that innovation? Complete it, please. The reader will better understand to the hardness achieved.
  • lines 139-140: please, do not use this sentence, it is very simple and it does not belong to a scientific paper of a higher rank (did it remain there by mistake from the diploma thesis?)

Author Response

We would like to thank the Reviewers for their comments and efforts towards improving our manuscript. All changes in the text are marked in blue.

line 27: It is such an unusual formulation, instead of "[5] He" it should be put there that Schwab [5] compared different methods ...

[MS]line 27 – corrected

lines 53-55: please describe in more details the construction of both composites: 1. about "single-layer HDF floor panels" we know its thickness and density, but we do not know the type of HDF surface treatment, the predominant type of wood (woods) in its structure and the method of its pressing due to the possible higher density in the surface layer (density profile); 2. about "double-layer floor panels made of innovative laminated wood materials with a rolled-up structured surface layer under the trade name Studio loftTM" we suspect that birch veneer was pressed with spruce veneer, but what is that innovation? Complete it, please. The reader will better understand to the hardness achieved.

[MS] line 53-55 – I supplemented the description to samples H, and F (HDF and Studio Loft) (I added the columns of table 1, changed the text above and below the table)

lines 64-65: the second parenthesis is missing ")"

line 67: There is no completed sentence - and ... what?

[MS]line 64-65 – corrected (text changed due to other comments from reviewers)

[MS]line 67 – correted (deleted „and”)

lines 139-140: please, do not use this sentence, it is very simple and it does not belong to a scientific paper of a higher rank (did it remain there by mistake from the diploma thesis?)

[MS]lines 139-140 – sentence was removed

Sincerely,

MS

Reviewer 2 Report

To the authors: This manuscript needs improvement. However, it represents a usually overlooked property. Good job. Almost there. 

L22-23: I recommend rewriting all these lines.

L27: He? Is that correct?

L30: Make a new paragraph in "For Brinnell"

L47: Please, state hypothesis, overall goal, and specific objectives.

I recommend breaking up this intro into four paragraphs

L53: HDF...spell it out.

samples of different structures.... Ok, I get it. But, how about let your reader know width and height?

I think the authors should be more careful when describing "structures".

L60: What RH and temperature conditions?

L74: This is confusing. (fig 1c?). Also, make a table or describe this in text format.

L84-87: Why did you determine these steps?

L107: Never assume that everybody knows the formula you are writing.

L133: Why not make a regression analysis (hardness vs density)? Also are the methods comparable? If so, why not an ANOVA?

Figure 8: The authors presented a nice graph with regression. While I am an entusiast of statistical analysis, I didn't see
any reference in the methods.

By this far, I don't if we are comparing among species designations. The manuscript has potential, but needs to be clearer.

L214: There are statistical methods to infer if something can be divided into groups. That would be K-mean clustering?

Several of these points seem to be discussion.

Author Response

We thank the referee for the careful and insightful review of our manuscript. All changes in the text are marked in blue. We address all of the concerns of the referee below.

L22-23: I recommend rewriting all these lines.

L27: He? Is that correct?

L30: Make a new paragraph in "For Brinnell"

[MS] L22-23: corrected

[MS] L27 corrected

[MS] L30 corrected

[MS] L47 At the end of the "introduction", the main purpose of the article was given (there is also a hypothesis in it). The specific goals are not specified. In our opinion, this is not justified. Please consider our point of view.

L53: HDF...spell it out.

samples of different structures.... Ok, I get it. But, how about let your reader know width and height?

[MS] L53 Corrected
(the thickness of the samples was given in the article, other dimensions were considered as insignificant – all samples had dimensions 7×150×150 mm)

L60: What RH and temperature conditions?

L74: This is confusing. (fig 1c?). Also, make a table or describe this in text format.

[MS] L60 – information added

[MS] L74 – corrected

L84-87: Why did you determine these steps?

[MS]L84-87: The application of load used was different from that required by EN 1354 (1 kN reached in 15 s, maintained for 25 s and entirely released). As is known in the Brinell method, the diameter of the imprint should be in the 0,25D ≤ d ≤ 0,6D range. Therefore, a lower force than that recommended in the standard was used. As a result, the diameter of the imprints in the tests ranged from 2.29 mm for sample F (innovative plywood) to 4.97 mm for sample H (pine). In addition, the time of force exertion was lengthened to reduce the elastic component of deformation. (this information was added to the text of the article)

L107: Never assume that everybody knows the formula you are writing.

[MS]L107 – corrected

L133: Why not make a regression analysis (hardness vs density)? Also are the methods comparable? If so, why not an ANOVA?

Figure 8: The authors presented a nice graph with regression. While I am an entusiast of statistical analysis, I didn't see

any reference in the methods.

By this far, I don't if we are comparing among species designations. The manuscript has potential, but needs to be clearer.

L214: There are statistical methods to infer if something can be divided into groups. That would be K-mean clustering?

Several of these points seem to be discussion.

[MS]L133 – the text was supplemented with additional statistical analyzes (ANOVA was used, information was added in text).

Sincerely,

MS

Reviewer 3 Report

In my opinion, the research has been conducted by means of a methodology that is not correct, even if the idea is interesting and the sampling wide.

The focus of the manuscript is on a new method to determine Brinell hardness. Therefore, one would expect a comparison between the new method proposed and those currently used in practice. So, the new method should have been compared with that prescribed by a technical standard, for instance EN 1534, using the same load applied and procedure of application. No technical standard is mentioned in the manuscript instead.

In particular:

  • The application of load used (Rows 84-86 and Fig 2.c) is very different from that required by EN 1354 (1 kN reached in 15 s, maintained for 25 s and entirely released; 3 min shall pass prior to measure the diameter of indentation);
  • The load applied of 294 N appears too little to me. EN 1354 requires a load of 1000 N, so the load used in the manuscript is only the 29.4% of that set by this standard;
  • The load applied appears also limited to assess a proper recovery. Maybe a too high load should be avoided because it could repress the recovery, but, as said, 294 kN is only 30% of the load set by EN 1354.
  • I do not fully understand why the author used Eqs. 3 and 4 to determine the value of h and x. These equations require assessing the diameter of the indentation, which is subjected recovery [it is not only h that recovers, but the entire volume of indentation (hemisphere) is subjected to recovery]. To me, it seems much easier to set the new method using only x values of elastic recovery shown in Figure 7.
  • The comparison made with literature values (Tab. 4) can be misleading, since different methods were used (see the loads applied);
  • The statistical analysis is entirely missing. Differences between HB values (Fig. 5) shall be compared by a statistical analysis to determine if they are significant or not and to group the results based on statistical significant values.

To summarize, the method proposed/assessed has to many uncertainties or flaws, and its relation with methods currently used is not clear. As said above, the idea is original and the sampling wide, so maybe it is worth modifying the manuscript. Anyway, I cannot say how this can be done and if satisfying results can be achieved. Maybe the authors could use EN 1534 to measure HB and calculate it first according to the standard and then based on the elastic recovery x, but this requires an entire new testing.

Author Response

We thank the referee for the careful and insightful review of our manuscript. All changes in the text are marked in blue. We address all of the concerns of the referee below.

The focus of the manuscript is on a new method to determine Brinell hardness. Therefore, one would expect a comparison between the new method proposed and those currently used in practice. So, the new method should have been compared with that prescribed by a technical standard, for instance EN 1534, using the same load applied and procedure of application. No technical standard is mentioned in the manuscript instead.

In particular:

The application of load used (Rows 84-86 and Fig 2.c) is very different from that required by EN 1354 (1 kN reached in 15 s, maintained for 25 s and entirely released; 3 min shall pass prior to measure the diameter of indentation);

The load applied of 294 N appears too little to me. EN 1354 requires a load of 1000 N, so the load used in the manuscript is only the 29.4% of that set by this standard;

The load applied appears also limited to assess a proper recovery. Maybe a too high load should be avoided because it could repress the recovery, but, as said, 294 kN is only 30% of the load set by EN 1354.

[MS] The application of load used was different from that required by EN 1354 (1 kN reached in 15 s, maintained for 25 s and entirely released). As is known in the Brinell method, the diameter of the imprint should be in the 0,25D ≤ d ≤ 0,6D range. Therefore, a lower force than that recommended in the standard was used. As a result, the diameter of the imprints in the tests ranged from 2.29 mm for sample F (innovative plywood) to 4.97 mm for sample H (pine). In addition, the time of force exertion was lengthened to reduce the elastic component of deformation. (this information was added to the text of the article)

In our opinion, comparing the test parameters used in the article with the parameters in EN 1534 exceeds the scope of work

I do not fully understand why the author used Eqs. 3 and 4 to determine the value of h and x. These equations require assessing the diameter of the indentation, which is subjected recovery [it is not only h that recovers, but the entire volume of indentation (hemisphere) is subjected to recovery]. To me, it seems much easier to set the new method using only x values of elastic recovery shown in Figure 7.

[MS] In equations 3 and 4, the relationships resulting from the rectangular triangle were used. The variable “d” is the indentation diameter which was measured with a Dino-Lite AM4815ZT EDGE digital microscope.

The comparison made with literature values (Tab. 4) can be misleading, since different methods were used (see the loads applied);

[MS] The hardness values from the literature have been converted (information that a different load than that used in the literature was used). Please consider our point of view.

The statistical analysis is entirely missing. Differences between HB values (Fig. 5) shall be compared by a statistical analysis to determine if they are significant or not and to group the results based on statistical significant values.

[MS] the text of the article was supplemented with statistical analysis (ANOVA was used, information was added in text).

Sincerely,

MS

Round 2

Reviewer 2 Report

I'll give you an example of a tweaked intro. 

The hardness is an crucial wood mechanical propriety, mainly because it positively and negatively correlates with density and moisture content, respectively. It also depends on the anatomical direction and can vary by up to 50% within the same species. The Janka and Brinell are the two most popular methods for determining hardness of wood materials. Schwab compared different hardness methods for 16 wood species and concluded that the Brinell hardness test of flooring materials obtained the most reliable results. 

I would drop "the measurement of wood har...."

The authors claimed that there is a hypothesis in the last paragraph of the intro. Sincerely, I can't see that. I do see an overall goal and a specific objective. I consider your point of view. However, the authors perhaps should consider what they are looking for. In other words, what was the major question that led to this research. That's a hypothesis. 

Table 1. All Latin names need to be in italic.  

L85: Isn't Figure 1c Samples A to D? I think you meant Figure 2?

L100: Remove as is known and start with "In the Brinell method"

L131: According to Equation 1. 

L160 Statistica 13 software (City, Country)[of the software, not where you are from] also a reference is needed. Statistica was owned by StatSoft for several years, but not Dell has control over it. So, ref is needed. 

L160: The Analysis of Variance (ANOVA). 

Ok, great, the authors described statistics. However, for the ANOVA, please describe the treatment levels. And for the regression, was it used a linear model? I am not being picky. Nowadays, we have to make science reproducible. 

L161: The p = 0.05 inference was made after F-test?

Table 4. How do you justify a p-value so low and R2 and correlation coefficient somewhat weak?

Conclusions:

Dear authors, this is an advise: a conclusion too long will make you manuscript boring and nobody will want read it, because you are somewhat discussing your data there.  

I recommend changing this to a more concise conclusion

Author Response

Thank you for your more detailed comments. All new changes made to the manuscript are highlighted in green.

  1. "I'll give you an example of a tweaked intro (...)".  We changed intro
  2. “I would drop "the measurement of wood har....". Corrected (for this reason the reference "Morath 1938" has been removed).
  3. “The authors claimed that there is a hypothesis in the last paragraph of the intro (…)”. We added a hypothesis (We took it into account in abstract, introduction and conclusions).
  4. Table 1. All Latin names need to be in italic.” Corrected

L85: “Isn't Figure 1c Samples A to D? I think you meant Figure 2?”. Corrected

L100: “Remove as is known and start with "In the Brinell method"” Removed.

L131: “According to Equation 1.” Added

L160 “Statistica 13 software (City, Country)[of the software, not where you are from] also a reference is needed (…)” Added the reference to TIBCO

L160: “The Analysis of Variance (ANOVA).” Corrected

  1. “Ok, great, the authors described statistics. However, for the ANOVA, please describe the treatment levels. And for the regression, was it used a linear model? I am not being picky. Nowadays, we have to make science reproducible.” W added a table 3 and its description in text. Additionally, we have changed Fig. 5 to make it easier to interpret.

L161: “The p = 0.05 inference was made after F-test?.” Yes

“Table 4. How do you justify a p-value so low and R2 and correlation coefficient somewhat weak?” It seems that this is due to the specificity of wood (from the wide range of analyzed wood materials). A broader attempt at interpretation has been included in the text, in the description of Table 5.

  1. “Conclusions: Dear authors, this is an advise: a conclusion too long will make you manuscript boring and nobody will want read it, because you are somewhat discussing your data there. I recommend changing this to a more concise conclusion” We have divided the conclusions into two parts. In our opinion conclusions formulated in this way, will be easier to interpret.

All comments were taken into account as much as possible. Thank you for your detailed comments. Sincerely, MS (corresponding author)

Reviewer 3 Report

I appreciate the effort of the authors in improving the manuscript, but I still do not agree with the setup of the study. In my opinion, two issues remain unchanged after the review:

- authors say that comparison with EN 1354 is out of scope. As for me, I do not see the point of investigating different/new methods to measure Brinell hardness without comparing them with well-established methods used in practice and standardized by technical standards.

- authors say that "As is known in the Brinell method, the diameter of the imprint should be in the 0,25D ≤ d ≤ 0,6D range". Still, EN 1534 does not refer to any min/max diameter of indentation, so the load applied in the study still appears too limited to me.

Author Response

Thank you for the broader justification of your comments. Taking advantage of the opportunity, and as a result of the comments of the second reviewer, we introduced a number of corrections and additions to the article (in responding to the comments from the second round, comments from the first round were also taken into account). All new changes made to the manuscript are highlighted in green.

  1. “- authors say that comparison with EN 1354 is out of scope. As for me, I do not see the point of investigating different/new methods to measure Brinell hardness without comparing them with well-established methods used in practice and standardized by technical standards.”
  2. “- authors say that "As is known in the Brinell method, the diameter of the imprint should be in the 0,25D ≤ d ≤ 0,6D range". Still, EN 1534 does not refer to any min/max diameter of indentation, so the load applied in the study still appears too limited to me.”

We had no intention of introducing a new Brinell hardness measurement method. As the reviewer rightly noted, there is a well-established engineering method of measurement described in EN 1534. The aim of the article was to compare the recovery capacity of a wide range of flooring materials. For this reason, among others, we used a lower than the standard force loading the measuring ball. The main motivation for not exceeding the imprint range (0.25D ≤ d ≤ 0.6D) was the desire to remain within the elastic range of the material, i.e. the desire to destroy the wood structure under the ball as little as possible. And this regardless of the hardness of the material. Thanks to this approach, very soft material (pine) and very hard (vertical plywood) were tested with the same parameters. In other words: if a force of 1 kN were used, there would be a risk that the impression diameter in sample "H" would reach a value close to that of the measuring ball.

Please consider that there is a method for determining empirical relationships between tensile strength and Brinell hardness (described, among others, in the publication https://doi.org/10.1533/9780857097576.479, page 508; a similar recommendation is in other sources that I do not quote here). This publication describes this imprint diameter limit recommendation.

Please take our view that diameters of the indentations resulting from the all tests should be within a range of 0.25D to 0.6D. The same parameters should be used for all tested materials, otherwise it would be impossible to verify the assumed hypothesis.

Thank you for your detailed comments. All comments were taken into account as much as possible. Some comments was not fully not included. These valuable comments will certainly be taken into account when continuing the research (I mean the comments from the first round of the review and the second). Sincerely, MS (corresponding author)